# Antioxidant Activity of Biogenic Cinnamic Acid Derivatives in Polypropylene

**DOI:** 10.3390/polym15173621

**Published:** 2023-09-01

**Authors:** Jannik Mayer, René Steinbrecher, Elke Metzsch-Zilligen, Rudolf Pfaendner

**Affiliations:** 1Division Plastics, Fraunhofer Institute for Structural Durability and System Reliability LBF, Schlossgartenstraße 6, D-64289 Darmstadt, Germany; jannik.mayer@lbf-extern.fraunhofer.de (J.M.); elke.metzsch-zilligen@lbf.fraunhofer.de (E.M.-Z.); 2Department Chemistry, University of Potsdam, Karl-Liebknecht-Str. 24-25, House 25, D-14476 Potsdam, Germany; rene.steinbrecher@iap-extern.fraunhofer.de

**Keywords:** polypropylene, stabilization, biogenic building blocks, *p*-hydroxycinnamic acid

## Abstract

Antioxidants (AOs) from natural resources are an attractive research area, as petroleum-based products can be replaced in polymer stabilization. Therefore, novel esters based on the *p*-hydroxycinnamic acids *p*-coumaric acid, ferulic acid and sinapic acid were synthesized and their structure properties relationships were investigated. The structures of the novel bio-based antioxidants were verified using NMR and Fourier-transform infrared (FTIR) spectrometry. The high thermal stability above 280 °C and, therefore, their suitability as potential plastic stabilizers were shown using thermal gravimetric analysis (TGA). The radical scavenging activity of the synthesized esters was evaluated by using the 2,2-diphenyl-1-picrylhydrazyl (DPPH) assay. Stabilization performance was evaluated in polypropylene (PP) using extended extrusion experiments, oxidation induction time (OIT) measurements and accelerated heat aging. In particular, the sinapic acid derivative provides a processing stability of PP being superior to the commercial state-of-the-art stabilizer octadecyl 3-(3,5-*di*-*tert*-butyl-4-hydroxyphenyl)propionate.

## 1. Introduction

Due to their versatile material properties, low price and remarkable processability, commodity polymers, such as low-density polyethylene (LLDPE) and PP, have a wide range of applications from food packaging to engineering applications [1,2,3]. The influences of heat, light, metal ions and mechanical stress lead to oxidative degradation during their processing and service life, resulting in the loss of chemical and physical properties like changes in the molar mass or a decline in the mechanical properties [4,5].

To retard the oxidative aging and protect the polymers against the harmful effects of oxidation, antioxidants are employed [6]. Antioxidants are divided into the two groups of primary and secondary antioxidants according to their reaction mechanism. Primary antioxidants scavange alkyl and peroxy radicals, which are formed via thermo-oxidative degradation. Secondary antioxidants, such as organosulfur compounds or phosphites, react with the intermediately formed hydroperoxides and convert them into non-reactive, thermally stable, harmless products [4,7]. The very effective sterically hindered phenols are among the most commercially used primary antioxidants. Typical representatives of this substance class in industrial applications are octadecyl 3-(3,5-*di*-*tert*-butyl-4-hydroxyphenyl)propionate (AO-1076) and pentaerythritol *tetrakis*(3,5-*di*-*tert*-butyl-4-hydroxyhydrocinnamate) (AO-1010) [8]. According to their synthesis based on 3-(3,5-*di*-*tert*-butyl-4-hydroxyphenyl)methylpropionate as the raw material, most of the stabilizers commercially used today have a petrochemical origin [9,10]. In addition, there are various concerns about the potentially harmful interactions of these structures with human metabolism, especially with regard to applications in the food sector [9,11,12]. Furthermore, these structures were shown to leach from the polymer matrix and demonstrably contaminate the environment [13,14,15].

In view of the described issues and the sustainability aspect, bio-based antioxidants are of great academic and economic interest. Well-known examples of biogenic plastic stabilizers are α-tocopherol [16,17], lignin [18,19,20], quercetin [21,22], dihydromyricetin [23,24], curcumin [25,26] and rutin [21]. However, the use of naturally extracted phenols is associated with some drawbacks. They tend to produce discoloration [21,22,25,27,28] and have often only limited temperature stability, which causes their degradation during polymer processing [28,29]. Furthermore, many natural phenols exhibit only low solubility in apolar polymers, such as PP and polyethylene (PE) [27,28,30,31].

One way of counteracting these problems is the synthetic adaptation of biogenic building blocks for their use in plastics. Zheng et al. were able to synthesize a thermally stable polymer using an enzyme-catalyzed polymerization of pyrogallic acid, which provides a greater thermo-oxidative stability of PP than various commercial stabilizers, such as AO-1010 or butylated hydroxyanisole (BHA) [32]. The tannin hexanoate and tannin hexanoate acetate esters synthesized by Grigsby et al. also showed a high UV-stabilizing effect in PP but contributed significantly less to the oxidative stability of PP than AO-1010 [33]. The esters of rosmarinic acid synthesized by Doudin et al. showed significantly greater thermal stability compared with the pure acid. In particular, the stearyl rosmarinate was found to be a strong radical scavanger according to the 2,2-diphenyl-1-picrylhydrazyl (DPPH) assay and shows a high melting temperature, as well as a long-term heat stabilizing effect in PE and PP [34]. Furthermore, Reano et al. described the synthesis of *bis*- and trisphenols based on the biogenic building block ferulic acid via enzymatically catalyzed transesterification reactions. The synthesized polyphenols show a greater contribution to the thermo-oxidative stability of the polybutylene succinate (PBS) than the commercial AO-1010 [35]. With the aim of expanding the library of known biogenic antioxidants and gaining a better understanding of structure–property relationships, we focused our development of new plastic stabilizers not only on ferulic acid but also on sinapic acid and coumaric acid as biogenic *p*-hydroxycinnamic acids. Thus, we report on the synthesis of *p*-hydroxycinnamic acid stearyl ester derivatives based on these three biogenic building blocks, which are present in many plants and can be isolated through biotechnological processes [36,37,38,39]. According to Figure 1, the structures were prepared using a simple two-step synthesis route. They are characterized by their high thermal stability and, as a result of the aliphatic stearyl functionality, high compatibility with the PP matrix. The antiradical properties of the ester structures presented in Table 1 were examined using the DPPH assay. Furthermore, the structure–property relationships were determined with respect to the degree of methoxy-group substitution on the phenolic ring (methoxylation) and the process, as well as long-term heat-stabilizing properties in PP. Since the new structures were tested as plastic stabilizers for the first time, they were compared with the commercially available α-tocopherol and AO-1076 to classify their antioxidant efficacy. While α-tocopherol is a well-known biostabilizer, AO-1076 is obtained from phenol as a petrochemical building block. Both structures are shown in Table 1. The results of this work provide essential insights for the future synthetic design of new biogenic stabilizers based on *p*-hydroxycinnamic acids.

## 2. Materials and Methods

### 2.1. Materials

Coumaric acid, stearyl alcohol, dibutyltin oxide (DBTO), DL-α-Tocopherol and 2,2-diphenyl-1-picrylhydrazyl (DPPH) were purchased from Alfa Aesar (Haverhill, MA, USA). Ferulic acid and concentrated sulfuric acid (98%) were purchased from Sigma-Aldrich Corp (St. Louis, MO, USA). Dichloromethane (DCM), silica gel (0.2–0.5 mm) and ethanol were purchased from Merck KGaA (Darmstadt, Germany). Methanol was purchased from VWR International, LLC. (Radnor, PA, USA). Sinapic acid was purchased from Apollo Scientific (Stockport, UK). Tonsil^®^ Optimum 210 FF bleaching earth was kindly supplied by Clariant AG (Muttenz, Switzerland). AO-1076 (octadecyl 3-(3,5-*di*-*tert*-butyl-4-hydroxyphenyl)propionate) was received from BASF SE.

PP Moplen HF501N (MVR = 10 g 10 min^−1^, 230 °C, LyondellBasell Industries N.V., London, UK) and Moplen HP500N (MVR = 12 g 10 min^−1^, 230 °C/2.16 kg, LyondellBasell Industries N.V.) were purchased from Ultrapolymers (Lommel, Belgium). The polymers contained a small amount of basic stabilization of less than 400 ppm phenolic AO.

### 2.2. Synthesis Procedure

#### 2.2.1. Synthesis of the Coumaric Acid Methyl Ester

Coumaric acid methyl ester was synthesized according to Percec et al. [40]. A total of 5.00 g (30.46 mmol, 1.00 eq.) coumaric acid was dissolved in 100 mL (2465.67 mmol, 80.95 eq.) methanol in a single-neck flask with a reflux condenser under slight heating. A catalytic amount of concentrated sulfuric acid was then added and the reaction solution was heated to reflux overnight. After completion (reaction control by DC, running medium ethyl acetate/n-hexane 3:1), the reaction mixture was cooled to room temperature, and the methanol was completely removed under reduced pressure. The residue was taken up in ethyl acetate and washed with saturated NaHCO_3_ solution and water. The organic phase was dried over sodium sulfate and the solvent was removed under reduced pressure. The residue was dried in vacuo and used for the subsequent reaction without further purification. A total of 5.13 g of a colorless solid was obtained (yield 95%). 

#### 2.2.2. Synthesis of the Ferulic Acid Methyl Ester

Ferulic acid methyl ester was synthesized according to Masuda et al. [41]. In a single-neck flask, 10.00 g (51.50 mmol, 1.00 eq.) ferulic acid was dissolved in 200 mL (4931.34 mmol, 95.75 eq.) methanol under slight heating. Subsequently, a catalytic amount of concentrated sulfuric acid was added. The yellow solution was heated at 50 °C for 3.5 h (reaction control by DC, running medium ethyl acetate/n-hexane 3:1). After cooling to room temperature, the reaction mixture was poured into 250 mL chloroform. The reaction mixture was washed with water and saturated NaHCO_3_ solution. The organic layer was dried over sodium sulfate and the solvent was finally removed at the rotary evaporator. After drying in a high vacuum, 10.10 g of a slightly yellowish oil was obtained (yield 94%).

#### 2.2.3. Synthesis of the Sinapic Acid Methyl Ester

Sinapic acid methyl ester was synthesized according to Fujita et al. [42]. A total of 10.00 g (44.60 mmol, 1.00 eq.) sinapic acid was dissolved in 140 mL (3451.94 mmol, 77.40 eq.) methanol under slight heating. Subsequently, a catalytic amount of concentrated sulfuric acid was added and heated for 7 h to reflux. After cooling, the reaction solution was stirred overnight at room temperature. Afterward, the reaction mixture was cooled in the refrigerator for 1 h. The precipitated colorless needles were filtered, washed with methanol and dried overnight in a vacuum drying oven at 50 °C. A total of 7.35 g of a colorless solid was obtained (yield 69%).

#### 2.2.4. Synthesis of the Cinnamic Acid Stearyl Esters

The synthesis of the stearyl ester derivatives was carried out via a bulk transesterification route according to Fischer et al. [43]. The cinnamic acid methyl ester (1.00 eq.) and stearyl alcohol (1.03 eq.) were placed in a dried Schlenk flask with a condensation bridge and attached cold trap. The reaction mixture was melted at 120 °C and degassed three times. DBTO (0.04 eq.) was added to the melt and the reaction mixture was heated to 135 °C at a pressure of 200–800 mbar. Reaction control was performed with ^1^H NMR spectroscopy by observing the decreasing signal of the methyl ester group at 3.80 ppm. The required reaction time varied strongly between the different cinnamic acid methyl esters. While the coumaric acid ester achieved conversion above 80% after 2 h, the esters of ferulic and sinapic acid required 14 and 18 h, respectively, for the same conversion. After completion of the reaction, the temperature was increased to 155 °C and the pressure was gradually reduced to 1 × 10^−3^ mbar to remove the excess stearyl alcohol from the reaction mixture. The vacuum was broken by adding nitrogen and the organic melt was cooled down to room temperature. The solidified melt was dissolved in DCM, bleaching earth was added and the reaction mixture was heated to reflux for 30 min. Subsequently, the reaction mixture, cooled again to room temperature, was filtered through a glass frit covered with silica gel. DCM was removed in a vacuum and the obtained product was dried in a vacuum drying oven. It can be recrystallized from ethanol if necessary.

Colorless fine powder of coumaric acid octadecyl ester (yield: 82%). T_m_ = 94 °C.

^1^H NMR (300 MHz, Chloroform-d) σ = 7.65 (d, 1 H, -C_arom_-CH=CH-COO-), 7.44 (d, 2 H, -C_arom_-H), 6.86 (d, 2 H, -C_arom_-H), 6.33 (d, 1 H, -C_arom_-CH=CH-COO-), 5.62 (s, 1 H, OH), 4.19 (t, 2 H, -COO-CH_2_-C_17_H_35_), 1.70 (m, 2 H, -COO-CH_2_-CH_2_-C_16_H_33_), 1.26 (m, 30 H, -COO-CH_2_-CH_2_-C_15_H_30_-CH_3_), 0.88 (t, 3 H, -COO-CH_2_-CH_2_-C_15_H_30_-CH_3_) ppm. ^13^C NMR (76 MHz, Chloroform-d) σ = 168.33, 158.35, 144.98, 130.15, 127.06, 116.09, 115.41, 65.07, 32.07, 29.84–29.43, 28.87, 26.12, 22.83, 14.25 ppm. IR (ATR, ν) = 3370 (OH), 2919 (CH_2_), 2847 (CH_3_), 1660 (C=O), 1602 (C=C), 1273 (C-O-C) cm^−1^.

Colorless fine powder of ferulic acid octadecyl ester (yield: 82%). T_m_ = 67 °C.

^1^H NMR (300 MHz, Chloroform-d) σ = 7.63 (d, 1 H, -C_arom_-CH=CH-COO-), 7.06 (m, 2 H, -C_arom_-H), 6.93 (d, 1 H, -C_arom_-H), 6.32 (d, 1 H, -C_arom_-CH=CH-COO-), 5.92 (s, 1 H, OH), 4.19 (t, 2 H, -COO-CH_2_-C_17_H_35_), 3.92 (s, 3 H, -O-CH_3_), 1.70 (m, 2 H, -COO-CH_2_-CH_2_-C_16_H_33_), 1.26 (m, 30 H, -COO-CH_2_-CH_2_-C_15_H_30_-CH_3_), 0.88 (t, 3 H, -COO-CH_2_-CH_2_-C_15_H_30_-CH_3_) ppm. ^13^C NMR (76 MHz, Chloroform-d) σ = 167.52, 148.07, 146.91, 144.30, 127.20, 123.16, 115.82, 114.86, 109.46, 64.76, 56.08, 32.06, 29.84, 28.92, 26.14, 22.82, 14.24 ppm. IR (ATR, ν) = 3387 (OH), 2916 (CH_2_), 2850 (CH_3_), 1714 (C=O), 1635 (C=C), 1263 (aryl-O), 1155 (C-O), 1031 (O-CH_3_) cm^−1^.

Colorless fine powder of sinapic acid octadecyl ester (yield: 82%). T_m_ = 78 °C.

^1^H NMR (300 MHz, Chloroform-d) σ = 7.55 (d, 1 H, -C_arom_-CH=CH-COO-), 6.77 (s, 2 H, -C_arom_-H), 6.32 (d, 1 H, -C_arom_-CH=CH-COO-), 5.78 (s, 1 H, OH), 4.19 (t, 2 H, -COO-CH_2_-C_17_H_35_), 3.91 (s, 6 H, -O-CH_3_), 1.67 (m, 2 H, -COO-CH_2_-CH_2_-C_16_H_33_), 1.25 (m, 30 H, -COO-CH_2_-CH_2_-C_15_H_30_-CH_3_), 0.87 (t, 3 H, -COO-CH_2_-CH_2_-C_15_H_30_-CH_3_) ppm. ^13^C NMR (76 MHz, Chloroform-d) σ = 167.36, 147.35, 143.18, 137.22, 128.86, 117.07, 105.17, 63.10, 56.45, 34.12, 30.95, 27.66, 25.27, 22.81, 15.59 ppm. IR (ATR, ν) = 3521 (OH), 2917 (CH_2_), 2849 (CH_3_), 1706 (C=O), 1635 (C=C), 1282 (aryl-O), 1156 (C-O), 1103 (O-CH_3_) cm^−1^.

### 2.3. Methods and Characterization

#### 2.3.1. Structural Characterization

NMR spectra were recorded at room temperature with a Bruker NanoBay 300 spectrometer (Billerica, MA, USA). NMR chemical shifts were referenced relative to the used solvent. FTIR spectra were recorded by using a Nicolet 8700 FTIR spectrophotometer with a Golden Gate ATR unit from Thermo Fisher Scientific (Waltham, MA, USA). For every spectrum, 32 scans were performed. The spectral resolution was 4 cm^−1^, while the measurements were recorded in a range between 4000 and 400 cm^−1^.

#### 2.3.2. DPPH Antioxidant Assay

According to the procedure of Zhan et al., a DPPH stock solution was prepared from 36 mg DPPH in 100 mL ethanol [44]. By diluting the stock solution, the DPPH working solution used for the measurement could subsequently be prepared with a concentration of 91 µM. During the course of the measurement, 2.97 mL of this DPPH working solution was placed in a quartz cuvette from HELLMA ANALYTICS and the absorbance at 515 nm was determined in a 2600i spectrophotometer from Shimadzu (Kyoto, Japan) for 12 min with an accumulation time of 0.1 s. After a short measurement period, 30 µL of an ethanolic antioxidant solution with a concentration of 2.5 mM was added. The radical scavenging effect was calculated using [(A_0_ − A_2min_)/A_0_]·100%, where A_0_ and A_2min_ are the absorbance values before and 2 min after adding the ethanolic antioxidant solution to the DPPH solution, respectively.

#### 2.3.3. Characterization of AOs

OIT and thermogravimetric analysis (TGA) were carried out on a TGA-DSC1 from Mettler Toledo (Columbus, OH, USA). TGA measurements were performed in the range of 35 to 600 °C with a heating rate of 10 °C min^−1^ under nitrogen and synthetic air atmosphere. The measurements were carried out with sample quantities between 4.40 mg and 119.97 mg in 100 μL alumina crucibles from THEPRO GbR. OIT measurements were conducted according to the standard method (DIN EN ISO 11357-6:2013 [45]). First, the sample was equilibrated under a nitrogen flow at 50 mL min^−1^; then, the atmosphere was switched from nitrogen to synthetic air at the same flow rate. The oxidation of the samples was detected as a significant increase in the heat flow according to the exothermic character of the oxidation reaction. The obtained OIT was observed as the onset peak of the heat flow increase and was calculated by using STARe software Version 14.00. Measurements were carried out in duplicate. Differential scanning calorimetry (DSC) measurements were conducted with a DSC822 e (Mettler Toledo) in the range of −30 to 200 °C with a heating rate of 10 K min^−1^ in a nitrogen atmosphere. Alumina crucibles (40 μL) with lid from Mettler Toledo were used for the measurement.

#### 2.3.4. Microextruder Experiments: Determination of the Processing Stabilization Performance

The melt-stabilizing effect of the additives and their impact on the processing stability during extrusion was investigated by compounding the polymer and the additives in a DSM Xplore 5cc twin-screw microcompounder by Xplore Instruments BV (Sittard, The Netherlands). A total of 2985 mg PP (Moplen HF501N, LyondellBasell Industries N.V.) was combined with 15 mg of the corresponding additives and compounded at a set barrel temperature of 200 °C and a screw speed of 200 rpm for 30 min. During compounding, the vertical force was measured to examine the change in melt stability and properties. The synthesized antioxidants were compared with the state-of-the-art stabilizer, namely, octadecyl 3-(3,5-*di*-*tert*-butyl-4-hydroxyphenyl)propionate (AO-1076) and the commercially available bio-based antioxidant α-tocopherol. Measurements were carried out in duplicate.

#### 2.3.5. Preparation of PP Compounds for OIT and Mechanical Properties 

The PP samples based on Moplen HP500N from LyondellBasell Industries N.V. were produced by using a co-rotating twin-screw lab extruder (Process 11, Thermo Fisher Scientific) with extrusion temperatures of 200 °C. After cooling in a water bath, the polymer strand was granulated using a lab pelletizer (VariCut, Thermo Fisher Scientific). The compositions of the produced formulations are compiled in Table 2. The granules were used for the melt volume rate (MVR) and OIT measurements. The MVR was determined by using an MI-2 capillary rheometer (GÖTTFERT, Buchen, Germany) at 230 °C with a 2.16 kg load using a die with L/D = 8 mm/2.095 mm according to DIN EN ISO 1133-1 [46]. Furthermore, tensile test bars were injection molded using a Babyplast 6/10P mini injection molding machine from Christmann Kunststofftechnik GmbH (Kierspe, Germany). The melt temperature was 210 °C, while the temperature of the injection molding tool was 40 °C. According to DIN EN ISO 1133-1, the investigation of the mechanical properties of the injection molded test bars was performed with a zwickiLine Z2.5 universal testing machine (Zwick Roell, Ulm, Germany) in standard conditions (23 °C, 50% rel. humidity). To generate comparable material output values, the test bars were tempered for 30 min at 150 °C before determining the initial mechanical properties. In the course of oven aging at 150 °C, the elongation at break fell from 500% to 200% within 30 min and remained at this value until the thermo-oxidative degradation finally took place. The tempering effect was explained by the storage temperature just below the PP melting point of 165 °C [47]. Five tensile test bars were measured for each mechanical test.

## 3. Results and Discussion

### 3.1. Characterization of the Hydroxycinnamic Acid Stearyl Esters

The reaction control during the course of the transesterification reaction was carried out by observing the decrease in the methyl ester singlet at 3.80 ppm with simultaneous occurrence of the methylene triplet at 4.19 ppm in the ^1^H NMR spectra. Figure 1 shows an example of this triplet in the ^1^H NMR spectrum of the SinSa. Furthermore, the synthesized structures were characterized via ^13^C and IR spectroscopy. The spectra can be found in the Appendix A.

### 3.2. Thermal Stability of the Hydroxycinnamic Acid Stearyl Esters

Sufficient thermal stability of the synthesized hydroxycinnamic acid esters above 200 °C is a necessary requirement for incorporation into polyolefins via compounding processes. Figure 2 shows the TGA curves of the three synthesized cinnamic acid esters obtained under a nitrogen atmosphere. Furthermore, TGA data of the prepared structures are displayed in Table 3. While T_5%_ and T_10%_ are the temperatures at which the mass loss during heating is 5% and 10%, respectively, T_max_ is the temperature at which maximum mass loss takes place. This value is obtained from the DTG curves by deriving the corresponding TGA curve. The synthesized structures show significant thermal degradation only when close to 300 °C and are, therefore, suitable for incorporation into polyolefins.

### 3.3. Antioxidant Activity of the Hydroxycinnamic Acid Esters

Evaluation and classification of the antioxidant activity of the synthesized hydroxycinnamic acid derivatives was carried out via a DPPH assay. The UV-vis spectrum of the DPPH radical had an absorption peak at 515 nm. According to Lambert–Beer’s law, the absorption value of this peak decreased with decreasing DPPH concentration. Since structurally different H-donors scavenge the DPPH molecule at different rates, this method provides information on the antioxidant activity of the examined H-donors [44,48,49]. Figure 3 shows the radical scavenging effect for the phenolic stabilizers calculated from the decoloration kinetics curves. The DPPH assay shows that the sinapic acid derivative had the greatest antioxidant activity among the synthesized hydroxycinnamic acid esters as a result of the greatest degree of steric hindrance. As already reported in the literature, the electro-donating methoxy group (-O-CH_3_) increased the stability of the aryloxy radical formed after H abstraction. Due to the greater stability of the resulting aryloxy radical, the speed of the H transfer from the phenolic hydroxy group to the DPPH radical was significantly increased. This resulted in a stronger absorption decay at 515 nm and, therefore, a higher measured radical scavenging effect [50,51]. As is known from butylohydroxytoluene (BHT), the transfer of the phenolic hydrogen radical to the DPPH radical is impeded and significantly slowed down by the sterically demanding *ortho*-positioned *tert*-butyl groups on the aromatic moiety of the AO-1076 [49,52]. The highest antioxidant activity was exhibited by α-tocopherol, which is known as a very effective antioxidant and, therefore, often used as a reference for this method [53].

### 3.4. Performance as Processing Stabilizer

Using continuous extrusion in a microextruder, several extrusion cycles can be simulated to determine the process-stabilizing properties of the synthesized hydroxycinnamic acid derivatives. In the course of the measurement, the synthesized structures and PP were subjected to an extrusion circle for 30 min and the residual force was detected. The measured residual force depends on the viscosity of the melt, and thus, the molecular weight of the PP. The force retentions observed after 10 min, 20 min and 30 min are summarized in Table 4. The continuous extrusion time of 30 min corresponds to about 10 processing steps in a multiple extrusion experiment.

The blank sample PP0 shows a substantial drop in the force and, therefore, a significant degradation of the PP during the measurement period. The addition of the synthesized hydroxycinnamic acid stearyl ester and commercial state-of-the-art stabilizers, such as AO-1076 or α-tocopherol, prevents the oxidative degradation of the PP and reduces the drop in the detected forces. Among the hydroxycinnamic acid derivatives, the sinapic-acid-based ester SinSa had the greatest force retention, and thus, the greatest stabilizing effect. The higher substitution degree at the aromatic moiety with methoxy-groups led to a higher radical stability, and thus, a higher antioxidant activity of the SinSa due to the +M effect of the methoxy group [50,51]. The higher antiradical efficiency of the sinapic acid derivative ultimately resulted in the best melt-stabilizing properties among the synthesized hydroxycinnamic acid esters. The drop in the force measured in the extruder, and thus, the process-stabilizing effect of SinSa was comparable with the commercially available AO-1076. The greatest force retention after an extrusion time of 30 min and the best process-stabilizing effect were achieved by the addition of α-tocopherol, which is already known as an extremely effective process stabilizer [16,54].

### 3.5. OIT Measurements

Oxidation induction time (OIT) measurements are a well-established method to determine and evaluate the thermo-oxidative stability of polyolefins like PP. In the course of the short-term test carried out in a DSC, the time of the onset of oxidation of the sample under oxidative conditions is determined. Higher OIT values indicate a higher extended resilience of the polymer against thermo-oxidative degradation [55]. Since different phenolic stabilizers in the same polymer matrix show different OIT values, these results provide information on the efficiency of the added phenolic antioxidant to stabilize the polymeric material against thermo-oxidative degradation. Thus, the antioxidant activity of the phenolic structures and their ability to stabilize the PP can be compared [56,57]. Table 5 shows the determined OIT values. All samples were measured at T_M_ = 220 °C.

All samples containing the cinnamic acid stearyl esters show higher OIT values compared with the non-stabilized sample PP0. Due to having the highest antiradical activity among the synthesized structures, the sample containing the sinapic acid derivative had the highest OIT value. Nevertheless, the samples with the commercially available stabilizers AO-1076 and α-tocopherol show much higher OIT values, and thus, thermo-oxidative stabilities. The reason for the difference in the OIT values was probably the lack of regeneration mechanisms in the case of the hydroxycinnamic acid esters. It is known that the commercially used Methyl 3-(3,5-*di*-*tert*-butyl-4-hydroxyphenyl)propionate (metilox) derivatives, such as AO-1076, can regenerate a stabilizing phenol species via their saturated alkyl group in the *para*-position of the aromatic compound, forming a quinone methide. Therefore, these derivatives exhibit a superstoichiometric activity [58,59]. This also correlates with the results of Reano et al., whose described ferulic-acid-based *bis*- and trisphenols without double bond in the α,β-position of the aromatic show similar large OIT values in PP to the commercially available AO-1010 [35]. Similarly, Beer et al. found that the presence of such a conjugating double bond in the α,β-position to the phenolic unit results in lower OIT values in the case of their macromolecular antioxidants with bound sterically hindered phenols [60]. Moreover, the dimers and trimers formed as degradation products of α-tocopherol show a high-melt-stabilizing effect in PP and PE [54,61]. In the case of hydroxycinnamic acid esters, comparable regeneration mechanisms are not possible due to the presence of a double bond in the α,β-position to the phenolic moiety, which ultimately results in a reduced contribution to the oxidative stability of the PP [12]. Figure 2 shows a comparison of the possible regeneration mechanisms in the absence and presence of the double bond in the α,β-position.

Since there was no correlation between OIT results and the long-term heat stability of the material, the PP samples were additionally examined via accelerated oven aging [56,62]. The aging process and the degradation of the PP can be determined via MVR values and mechanical parameters. The MVR value is determined using the molecular weight of the polymer, as well as existing branching or cross-linking. The oxidative degradation and the associated chain scission results in a higher fluidity of the polymer melt and, therefore, higher MVR values [43,63]. Both the samples with the added hydroxycinnamic acid octadecyl esters and the sample with the α-tocopherol show a significant increase in the MVR value before reaching 400 h (Figure 4a). The much smaller contribution of α-tocopherol to the long-term thermal stability of polyolefins compared with sterically hindered phenols, such as AO-1076, has already been described in the literature [64,65]. It is assumed that the oxidation products of α-tocopherol are mainly effective in quenching alkyl radicals, but not peroxyl radicals [64]. Among the hydroxycinnamic acid esters, the ferulic acid and sinapic acid derivatives provide the greatest contribution to the long-term heat stabilization of PP. Due to their greater degree of methoxylation and their consequently higher antioxidant activity [50,51], the peroxy radicals formed in the course of the thermo-oxidative degradation process are more effectively scavenged by these two derivatives. The faster degradation of the PP compound PP3 with the sinapic acid ester derivative compared with the compound with the ferulic acid stearyl ester can presumably be attributed to the slower radical adduct formation due to the greater steric hindrance at the aromatic moiety of the SinSa. Thus, the formation of these adducts plays a crucial role in the anti-radical mechanism for both sinapic and ferulic acid [66]. Even though the ferulic acid and sinapic acid octadecyl esters outperformed α-tocopherol in terms of their long-term heat stabilizing effect, the AO-1076 shows the greatest stabilizing effect among all the investigated stabilizers. A significant decrease in chain length and, consequently, an increase in MVR values are observed above an aging time of 1560 h. The MVR values and results of the mechanical measurements are shown in the Appendix A. In view of the structural similarity between the hydroxycinnamic acid esters and the AO-1076, the presence of the double bond in the α,β-position to the phenolic moiety appears to be a critical factor for the long-term heat stabilization of the phenols. In contrast with AO-1076, the formation of the quinone methide and the regeneration of the stabilizing phenolic species cannot take place in the case of the synthesized derivatives, which ultimately results in their reduced long-term heat stability of the hydroxycinnamic acid esters (Figure 2). Also, the ferulic-acid-based *bis*- and trisphenols described by Reano et al. without a double bond in the α,β-position to the phenyl group provide significantly improved long-term heat stabilization in PBS, even though the test conditions were considerably milder at 40 °C. [35] Therefore, the removal of the α,β-unsaturation seems to be an opportunity to enhance the contribution of the *p*-hydroxycinnamic-based antioxidants to long-term heat stabilization.

Degradation and aging processes resulted in a reduction in the molecular weight, and thus, a deterioration in the mechanical properties of polymers like PP. Mechanical parameters such as elongation at break are, therefore, also suitable aging criteria and provide information on the degree of material damage at a particular point of the aging time [62,67]. The elongation at break results of the aged tensile bars are shown in Figure 4b and correspond to the MVR results. Thus, both samples with the synthesized hydroxycinnamic acids and α-tocopherol show a small contribution to the long-term thermal stability of the PP. Below an aging time of 550 h, a complete degradation, and thus, a decrease in the elongation at break of the PP test bars took place. The best stabilizing performance among the synthesized derivatives was again shown by the test bars with the ferulic and sinapic acid octadecyl ester (PP2 and PP3), which outperformed the stabilizing effect of α-tocopherol.

## 4. Conclusions

A simple method for the synthesis of hydroxycinnamic acid stearyl esters based on the biogenic building blocks coumaric acid, ferulic acid and sinapic acid is presented. The bio-based antioxidants were realized in a two-step synthesis and characterized using NMR and FTIR spectroscopy. The three esters featured high thermal stability above 280 °C and, therefore, are suitable for incorporation into plastics such as PP. The DPPH analysis and prolonged extrusion experiments show that the degree of substitution on the aromatic moiety was a critical factor in terms of anti-radical and melt-stabilizing properties. According to its highest methoxylation degree, the sinapic acid derivative provides superior anti-radical properties and a processing stability comparable with AO-1076, which is a commercially available and petrochemically based antioxidant. Due to the presence of the double bond in the α,β-position with the phenolic group and the resulting suppression of the regeneration mechanism, the hydroxycinnamic acid derivatives contributed less to the oxidative stability and long-term heat stabilization of the PP. To develop potent biogenic antioxidants with strong long-term heat stabilizing properties, future work should, therefore, focus on the preparation of the corresponding saturated *p*-hydroxycinnamic analogs lacking a double bond in the discussed α,β-position.

## 5. Patents

Mayer, J.; Metzsch-Zilligen, E.; Pfaendner, R. Use of substituted cinnamic acid esters as stabilizers for organic materials, stabilized organic material, method for stabilizing organic materials and specific cinnamic acid esters WO2021191078 (A1) 30 September 2021.

## Data Availability

The data presented in this study are available on request from the corresponding author.

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
