# Peer review of "Antioxidant Activity of Biogenic Cinnamic Acid Derivatives in Polypropylene"

_polymers, 2023, doi:10.3390/polym15173621_

Round 1

Reviewer 1 Report

The manuscript reports "Antioxidant activity of biogenic cinnamic acid derivatives in PP", there are some issues that need to be corrected before continue the publication process, following they are detailed;

-for citation of references in main text, I recommend to follow the Instructions for authors of Journal.

-In Materials  section indicate that 2 different PP resins were supplied, which of them was used in experimentation? I mean these 2 PP have different MVR which mean that are different.

-For characterization test it is needed to provide more details: which technique was used for FTIR? KBr plates, ATR, casting, liquid, etc?; for thermal analysis which was the sampel size and kind of pans used?;how many replies were done for mechanical tests?

there is a text in lines 255 256 that has not sense with manuscript.

-in figure 3, which is the variable that corresponds to X axis? i mean only indicate from 0 to 100 but which units?

-According with data in table 5, the OIT was too low for new antioxidants proposed, compared with commercials one, so why in conclusions indicate that the OIT measurements show a significant improved oxidation stability.

-in figure 3, which is the variable that corresponds to X axis? i mean only indicate from 0 to 100 but which units?

-line 353 please indicate which figure it is refered.

In figure 4a, what is it means thta lines go up to the image after last poiint? An d what about PP_c1 and C3? why they are not plotted in figures 4a and b? For figure 4b i recommend to use broad bars for a better identification.

-It is important that authors highlight the benefits of highlights for use of cinnamic acid derivatives as antioxidants additives in PP, due the comparative with commercials ones is too low the OIT measurements.

-The conclusions are poor and need to be improved highlighting the most important findings of works.

- I recommend to use the abbreviation for PP instead of write the word complete.

Reviewer 2 Report

This research paper explores the synthesis of new esters derived from p-hydroxycinnamic acids for use as antioxidants in polymer stabilization. These esters demonstrated high thermal stability and potent radical scavenging activity. Their effectiveness in stabilizing polypropylene outperformed a standard commercial stabilizer. However, the long-term heat stability provided by these bio-based antioxidants needs further improvement. The work thus presents a promising approach towards sustainable polymer stabilization

Here are some critical points to consider in the review of this paper:

Clarity and Organization: The paper seems clear and well-organized, presenting the aims, methods, results, and implications in a logical sequence. 

Originality: The study presents a novel approach in synthesizing esters based on p-hydroxycinnamic acids for polymer stabilization, which seems innovative. Yet, the authors should clearly state how their work contributes to the existing knowledge in this area. It would be beneficial if they compared their results with those from similar studies.

Methodology: The methods for synthesizing the esters and evaluating their properties appear to be well-described. However, the paper could provide more information about the conditions used during synthesis and evaluation, such as temperatures, reaction times, and solvents. These details are critical for repeatability.

Interpretation: The authors should clarify the relationship between the degree of methoxylation and the anti-radical efficacy. It would be also useful if they discussed why the hydroxycinnamic acid derivatives contribute to a lesser extent to the long-term heat stability of the plastics. Are there any strategies to improve this?

Terminology: The paper should ensure that non-specialist readers can understand it, providing definitions or explanations for specific terms (like "OIT measurements", "AO-1076", "methoxylation").

Conclusion and Future Work: The conclusion should be strong, summarizing the main findings and their implications. Furthermore, suggesting future studies to improve the long-term heat stability of the plastics with hydroxycinnamic acid derivatives could provide a clear direction for subsequent research.

Overall, this paper seems to present a valuable contribution to the field of polymer stabilization with bio-based antioxidants. However, additional details and clarifications are necessary for publishing

Reviewer 3 Report

MDPI - polymers-2512252

Antioxidant activity of biogenic cinnamic acid derivatives in polypropylene

This manuscript reports findings on structures of the novel bio-based antioxidants. NMR and FTIR techniques were used to characterize the structure of experimental material. TGA analysis showed the high thermal stability of biogenic cinnamic acid derivatives. In general, this study covers a number of useful characterization techniques to evaluate the research hypothesis. In my opinion the topic and research subject are interesting and has novelty enabling this manuscript to be published in Polymers. However, I would suggest following points to be considered by authors to improves the quality of this manuscript.

1)      Abstract has been written well; however following points can be considered to improve it:

·         future prospective (implication of the results).

·         Statistical evaluation for significant/non-significant differences in data. You need to quantitatively demonstrate results in this section with statistical evaluation.

2)      Please give a brief survey of literature about similar reports on incorporating natural antioxidants esters as potential plastic stabilizers in packaging materials. Then explain the novelty of your research compared to those literature.

3)      Also, provide the “research hypothesis” at the end of your introduction. English grammar in some parts needs to be improved.

Author Response

"Please see the attachment

Round 2

Reviewer 1 Report

After review the corrected version, I wish to thanks to authors for consider the comments/corrections done to previous version, now the manuscript shows a significant improve so now I can recommend to continue  the publication process